psychology

inhibition, appetite, ingestive behaviour, Western-style diet

**Author for correspondence:**
Richard J. Stevenson
e-mail: dick.stevenson@mq.edu.au

# Hippocampal-dependent appetitive control is impaired by experimental exposure to a Western-style diet

Richard J. Stevenson[1], Heather M. Francis[1], Tuki Attuquayefio[2], Dolly Gupta[1], Martin R. Yeomans[3], Megan J. Oaten[4] and Terry Davidson[5]

[1]Department of Psychology, Macquarie University, Sydney, New South Wales 2109, Australia
[2]Department of Psychiatry, Yale University, New Haven, CT, USA
[3]School of Psychology, University of Sussex, Brighton, UK
[4]School of Applied Psychology, Griffith University, Gold Coast, Australia
[5]Center for Behavioral Neuroscience, American University, Washington, WA, USA

RJS, 0000-0002-2461-5777

Animals fed a Western-style diet (WS-diet) demonstrate rapid impairments in hippocampal function and poorer appetitive control. We examined if this also occurs in humans. One-hundred and ten healthy lean adults were randomized to either a one-week WS-diet intervention or a habitual-diet control group. Measures of hippocampal-dependent learning and memory (HDLM) and of appetitive control were obtained pre- and post-intervention. HDLM was retested at three-week follow-up. Relative to controls, HDLM performance declined in the WS-diet group ($d = 0.43$), but was not different at follow-up. Appetitive control also declined in the WS-diet group ($d = 0.47$) and this was strongly correlated with HDLM decline ($d = 1.01$). These findings demonstrate that a WS-diet can rapidly impair appetitive control in humans—an effect that could promote overeating in consumers of a WS-diet. The study also suggests a functional role for the hippocampus in appetitive control and provides new evidence for the adverse neurocognitive effects of a WS-diet.

## 1. Introduction

In animals, an extensive literature suggests that even brief exposure to a Western-style diet (WS-diet), rich in saturated fat and added sugar, results in selective impairment on tests of hippocampal-dependent learning and memory (HDLM, e.g. [1–6]). Emerging evidence in humans suggests a similar conclusion (e.g. [7]).

While it is well established that much hippocampal processing relates to memory (e.g. [8–10]), its functions extend into many other domains including the regulation of appetite (e.g. [11]). Animal data suggest that hippocampal lesions impair the use of internal state to resolve appetitive-related decisions (e.g. [3,12,13]). Presumably, the same should occur in humans, with a WS-diet impairing hippocampal function, and thus relatedly, hippocampal-dependent appetitive control. This report aimed to test these predictions.

That a WS-diet affects human hippocampal function in young lean healthy people—paralleling animal studies who typically use young lean healthy rats or mice [14]—is a fairly recent finding. Early studies identified an association between consumption of a WS-diet, assessed by a validated food frequency measure, and performance on neuropsychological tests that are known to be dependent upon hippocampal function [15–17]. These studies included tests of verbal paired-associate learning, visual memory and episodic memory—with associations to a WS-diet sustained even after controlling for other likely sources of variability (e.g. exercise, mental and physical health etc.). These data suggest that the more a person conforms to a WS-diet, the more likely they are to perform poorly on hippocampal-dependent neuropsychological tasks. This correlational approach has been further augmented by the finding that hippocampal volumes on neuroimaging are also similarly correlated with a WS-diet, with this diet linked to smaller hippocampi in older samples [18,19]. A more crucial test, previously only examined in animals [14], is whether experimental exposure to a WS-diet actually causes hippocampal impairment. So far, only one study has reported this [7]. Here, young healthy lean participants were randomly assigned to undertake 4 days of either healthy control breakfasts or a week of Western-style breakfasts, high in saturated fat and added sugar—all consumed in a laboratory setting. Before and after the dietary exposure phase, all participants undertook control and hippocampal-dependent neuropsychological testing. The key finding of this study was that relative to controls, those in the Western-style breakfast group performed more poorly on one test of HDLM (Hopkins verbal learning task; but not on another, logical memory), while showing no change on control tests (forward and backward digit span). These data provide the first and as yet only demonstration in humans—paralleling animal studies—that brief exposure to a WS-diet causes impaired performance on a test of HDLM.

Current theories of appetitive motivation, most notably incentive salience theory, place particular emphasis on the differential neural basis of anticipatory (wanting) and consummatory (liking) reward, but are generally less focused on how the brain modulates these processes [20–22]. Such modulation is clearly important [22]. This is because animals and humans frequently encounter external cues associated with rewarding outcomes, and without some form of modulation behaviour would be at the whim of an ever-changing series of momentary desires. Modulation, therefore, allows prioritization of motivational goals, and one important means of achieving this is to draw upon internal bodily (i.e. interoceptive) state (e.g. [20], fig. 5). In the context of appetitive motivation, being sated should serve to reduce the reward value of externally perceived food cues (i.e. limiting approach) thereby controlling appetite.

Modulation in the incentive salience model can be achieved via direct dampening of brain reward-related systems. An alternative idea proposes that satiety controls appetitive behaviour by inhibiting the ability of excitatory food cues to activate associative networks connected with obtaining that food, which includes memories of its affective and sensory properties [3,12,13]. That is, satiety cues control appetitive behaviours by inhibiting the ability of excitatory food cues to activate their associative network. Within this framework, satiety cues function as diffuse contextual stimuli that modulate associative networks, including memories and behavioural outcomes, by enabling animals—and presumably people too—to anticipate when food cues will be followed by reinforcing outcomes.

Irrespective of which of these accounts of modulation might be correct, there are grounds for thinking that the hippocampus is involved in using interoceptive state to regulate appetite [11,23]—although noting that inhibitory effects in animals can be obtained seemingly independent of interoceptive state [24]. The hippocampus has an established experimental and theoretical history of involvement in inhibitory processing (e.g. [25–28]). Furthermore, the hippocampus receives extensive neurohormonal inputs concerning interoceptive state (e.g. [29]). The hippocampus could then serve either model under conditions where an excitatory food cue is perceived when sated. Under such conditions, the hippocampus could either inhibit retrieval of associative networks connected with that food and/or dampen activation of brain areas mediating reward. If the hippocampus becomes impaired, then such regulation should become less efficient. Consequently, perceiving food cues should trigger associative networks connected with obtaining that food, including memories of it and/or the activation of brain areas mediating reward, irrespective of interoceptive state. This will increase the likelihood of eating when full.

One set of findings that is consistent with this account of hippocampal function comes from human neuropsychology. HM [30], and other patients with hippocampal damage [31,32], all demonstrate

interoceptive and appetitive abnormalities. Several investigators noted that HM rarely reported feeling hungry or tired, even if testing had extended well beyond his usual meal schedule [33]. These observations led to a more formal test, evaluating both his interoceptive capacity and appetitive behaviour [30]. In this study, HM was offered a second main meal almost immediately after finishing his first. In addition, self-reports of hunger/fullness were made at various intervals before and after each meal. Not only did HM readily eat a second meal but he reported relatively little change in hunger/fullness. Although these findings were originally believed to reflect HM's concurrent amygdala damage (see [30]), subsequent animal studies have found that discrimination of hunger and satiety states is impaired following selective hippocampal lesions that leave the amygdala intact [34,35]. Together, this suggests that hippocampal damage is associated with impaired interoception and appetitive behaviour in both humans and animals [11,31,35].

A recent study attempted to more directly test the role of hippocampal-dependent processes in the control of appetite by interoceptive satiety cues [15]. Using a cross-sectional design, hungry participants who habitually ate either a healthy or a WS-diet, viewed familiar palatable snacks and reported how much they wanted to eat them. As the snacks were only viewed, these judgements rely upon memory. Next, after tasting each snack, participants rated liking for their flavour and whether they wanted more—with both of these ratings presumably more reliant on the actual sensory properties of the food. After a filling meal, participants were asked to repeat this wanting and liking test. Three findings emerged. First, snack wanting declined more across the meal than snack liking. This represents a form of appetitive control, with the snack judged as significantly less desirable (i.e. wanting) than it *actually* tastes (i.e. liking), thus presumably reducing the likelihood of further eating. Second, consumers of a WS-diet reported smaller changes in wanting relative to liking across the meal, than consumers of a healthy diet. This finding reflects a loosening of appetitive control. Third, the smaller the changes in wanting relative to liking (i.e. our index of appetitive control), the poorer the performance on a neuropsychological measure of HDLM, suggesting mediation of this effect by the hippocampus.

A significant limitation of Attuquayefio *et al.*'s [15] findings is that they are correlational. It is, therefore, not possible to conclude that a WS-diet causes impairment of HDLM, and relatedly, of appetitive control, as measured by the wanting and liking test. To test for causality, one would need to examine if experimental exposure to a WS-diet: (i) impairs HDLM; (ii) causes a similar pattern of responses on the wanting and liking test, as observed cross-sectionally in consumers of a WS-diet in Attuquayefio *et al.*'s [15] study; and (iii) that predicted outcomes (i) and (ii) correlate. This set of predictions, if confirmed, would imply that a WS-diet causes hippocampal impairment, and that one consequence of this is poorer appetitive control. More broadly, this would: (i) support the idea that hippocampally mediated processes underlie the ability of satiety cues to control appetitive behaviour; and (ii) that a WS-diet causes neurocognitive impairment that erodes appetitive control.

The experiment reported here set out to test these predictions, by randomly assigning young lean healthy participants who generally ate a good quality diet to either a Western-style (WS)-diet group, who consumed a WS-diet for one week or a control group who retained their habitual diet over this same period. All participants undertook an initial test session (Day 1) prior to any intervention, which included neuropsychological testing (HDLM and digit span) alongside various control measures to check that the groups were matched. Importantly, the wanting and liking test was completed before and after a filling breakfast. Participants then returned one week later following the end of the intervention phase (Day 8) and repeated the testing described for Day 1. They returned again approximately three weeks later for limited follow-up testing (Day 29), mainly to check if the consequences of the dietary intervention on HDLM were reversible—as suggested by recent animal data [36].

# 2. Method

## 2.1. Participants

Participants were recruited from Macquarie University either from the 1st-year participant pool for course credit or from campus-wide advertisements for a cash payment. All participants were screened prior to enrolment to ensure they were lean, healthy and currently consuming a nutritious diet (i.e. defined as a below-average score on a validated self-report measure of WS-diet; [37]—see Results for details). Exclusion criteria were: pregnancy, current/past metabolic, neurological or psychiatric

illnesses, food allergies, vegan/vegetarian, non-pork eater, currently dieting, recent significant diet-change, prescription medication use (other than the contraceptive pill and asthma medication), illicit drug use and current ill-health. Inclusion criteria were: aged 17–35, body mass index (BMI) between 17 and 26, below average WS-diet score, fluent English, and a depression, anxiety and stress scale (DASS; [38]) score below 25. Participants were allocated to experimental conditions based upon a pre-determined random schedule stratified by gender. This ensured approximately equal numbers of males and females in each group.

Based on prior work, we expected effect sizes (Cohen's $d$) of between 0.5 and 0.6 for the primary endpoints. To have an 80% chance of rejecting the null, with $d$ set at 0.55 and alpha at 0.05, we aimed to recruit 52 participants per group. In actuality, 110 completed Day 1, 105 Day 8 and 102 the follow-up on Day 29. Written consent was provided by each participant. The procedure was approved by the Macquarie University Ethics Committee, with full disclosure of the study aims on a debriefing following testing on Day 29.

## 2.2. Stimuli

### 2.2.1. WS-Diet group

On Days 1 and 8, these participants received a laboratory breakfast of a toasted sandwich and a milk shake, high in saturated fat and added sugar (total KJ = 4023; 33% fat [19% saturated], 51% carbohydrate [29% sugar] and 16% protein). On Days 2–7, participants were instructed to eat two Belgian waffles for breakfast or dessert on 4 days (total KJ for two waffles = 3376; 28% fat [15% saturated], 65% carbohydrate [31% sugar] and 7% protein), and to obtain a main meal and drink/dessert from a set of options from a popular fast-food chain on the other 2 days (total KJ per average meal/drink/dessert = 4127; 27% fat [15% saturated], 46% carbohydrate [25% sugar] and 27% protein). The waffles were provided to participants along with cash to obtain the fast-food meals. Setting aside these changes, participants were instructed to otherwise try and maintain their normal diet.

### 2.2.2. Control group

On Days 1 and 8, these participants received a laboratory breakfast consisting of a toasted sandwich and a milk shake, low in saturated fat and added sugar (total KJ = 2954; 13% fat [5% saturated], 29% carbohydrate [10% sugar] and 58% protein). On Days 2–7, they were asked to maintain their normal diet.

### 2.2.3. Wanting and liking test

Six breakfast foods were used: 10 ml of Coco pops in 10 ml of milk; 10 ml of Frosties in 10 ml of milk; 10 ml of Froot loops in 10 ml of milk; Mini-toast (crispy toasted bread squares approx. 3 by 3 cm) with jam; Mini-toast with Vegemite (a savoury spread); and Mini-toast with Nutella.

### 2.2.4. General ratings

These consisted of seven line scales (anchors: Not at all, Very) labelled: hunger, thirst, fullness, happiness, sadness, relaxedness and alertness. Mood/arousal ratings did not reveal any group-related differences and so are not further reported (noting these raw data are included in the electronic supplementary material).

## 2.3. Procedure

### 2.3.1. Day 1

All participants reported having fasted overnight as instructed. Biographical and health data were collected first, with participants completing measures of: physical activity [39], sleep, WS-diet [37], eating attitudes [40] and mood [38]. Height, weight and waist circumference were collected on each test day, but as these did not differ across days, only Day 1 data for these anthropometric variables are presented. Participants undertook a point-of-care diagnostic test for blood glucose (SensoCard), using a blood sample from a finger-prick device (Unistik), before (20 min) and after (10 min) breakfast. A urine sample was also obtained to assess biological correlates of the predicted neuropsychological changes, but these samples have yet to be processed by the test laboratory.

The Hopkins verbal learning test (HVLT) acquisition phase was then completed. The HVLT has multiple parallel forms of equal difficulty, three of which were counterbalanced across test Days 1, 8 and 29. The HVLT involves an initial set of three acquisition trials, followed after 20 min by a delayed recall trial [41]. The variable of interest was per cent retention, as for Attuquayefio et al. [7]. The HVLT has adequate reliability, and performance on this test is associated with hippocampal integrity (e.g. [41–43]). This was followed by forward digit span, which was not expected to be affected by the intervention. Digit span from the RBANS was chosen as it has multiple alternate forms and adequate reliability [44]. The reported RBANS score reflects performance across multiple trials of increasing difficulty (i.e. more digits to remember) rather than digit span per se.

The pre-breakfast wanting and liking test was then completed, using the same rating scale format as on all previous uses, to ensure procedural consistency (i.e. [15,45,46]). Participants received the six test foods in counterbalanced order. For each food sample, they were asked to rate on a line scale how much they would like to eat it now (anchors: Not at all, Very). They were then presented with each food again, but this time they were asked to consume it, and rate how much they liked it (scale as above) and how much more of it they would like now (anchors: None, A lot).

Participants then undertook a modified form of matrix reasoning (i.e. using a set time limit per trial on a subset of just the harder trials) as a proxy measure of intelligence, followed by the delayed test of the HVLT. Participants then completed the general ratings, followed by breakfast, where participants were instructed to eat as much as they could in the 10 min given for this meal (with ad libitum access to water). All uneaten food was collected for later weighing.

After a further set of general ratings, and the post-breakfast wanting and liking test (identical to the pre-breakfast test), participants evaluated the breakfast (for liking and fillingness, using line scales). The WS-diet group were then instructed about the diet intervention and on how to complete a diet diary. The diet diary was obtained on one day when eating waffles, and another day when eating a fast-food meal. Participants were also asked to record: (i) the days they ate the waffles and fast-food meals; (ii) how much they ate; (iii) photograph with their phone each eating bout with the waffles; and (iv) obtain receipts for the fast-food meals. Controls were asked to keep a food diary on one weekday and one weekend day.

### 2.3.2. Day 8

This was procedurally identical to Day 1, except matrix reasoning was replaced with two questionnaires—an adapted form of the everyday memory scale ([47]; referring to just the last week) and a food craving measure (as there were no group effects for either of these measures they are not further reported). Food diaries and related materials were collected and participants were asked about the occurrence of any adverse life events since Day 1. None were reported.

### 2.3.3. Day 29

Testing was conducted in the morning with no restrictions on prior food intake. The HVLT and digit span tests were completed. This was followed by the standard form of the everyday memory test and a food cravings measure (again as there were no group effects for either variable these measures are not further reported). General ratings were obtained, followed by the delayed test of the HVLT. A debriefing then followed about the specific purpose of the experiment.

## 2.4. Analysis

Scores from each rating scale type on the wanting and liking test were averaged across food type. This averaging was undertaken as the test is designed to assess wanting and liking for palatable foods, not specific individual foods. In addition, this analysis approach has been adopted with this test in previous studies, ensuring consistency of approach (i.e. [15,45,46]). The averaging procedure was conducted separately for the pre- and post-breakfast tests on both Days 1 and 8. As certain post-breakfast test ratings contained zero values, these data were non-normal and not amenable for transformation. Consequently, we calculated difference scores (pre–post breakfast) for each test day and for each rating scale type, and used these values in the analysis.

HVLT retention data were non-normal so bootstrapping was used. Diary data were analysed using FoodWorks 8 software (noting that diet diaries were not returned by one participant), and then intakes were averaged across the two record days. Data were analysed using independent t-tests, univariate ANCOVAs and mixed-design ANOVAs. For brevity, only p-values are reported for the non-primary outcomes.

**Table 1.** Descriptive statistics (M and 95% CI) for measures obtained on Day 1, with no differences between groups (all $p$'s > 0.23).

| variable | WS-diet group | control group |
|---|---|---|
| gender ($n$ = female/male) | 33/21 | 30/21 |
| age | 22.0 (20.8–23.1) | 22.6 (21.4–23.7) |
| body mass index | 22.0 (21.4–22.7) | 21.7 (21.1–22.3) |
| waist circumference (cm) | 77.8 (74.9–80.6) | 77.4 (75.4–79.3) |
| mood (DASS[a] total score) | 10.4 (8.3–12.6) | 10.5 (8.4–12.5) |
| Western-style diet score | 50.7 (48.1–53.2) | 52.9 (50.1–55.6) |
| activity (IPAQ[b] Met min) | 9121.5 (5930.2–12312.8) | 8166.2 (3963.4–12638.9) |
| TFEQ[c] restraint | 8.3 (7.2–9.5) | 8.0 (6.8–9.3) |
| TFEQ[c] disinhibition | 5.7 (4.9–6.5) | 5.9 (5.0–6.8) |
| TFEQ[c] hunger | 5.5 (4.4–6.5) | 5.8 (4.8–6.8) |
| sleep quality score | 4.3 (3.9–4.7) | 4.4 (4.0–4.7) |
| modified matrix reasoning (IQ) | 6.3 (5.9–6.8) | 6.7 (6.1–7.2) |

[a]Depression, anxiety and stress scale.
[b]International physical activity scale.
[c]Three factor eating questionnaire.

# 3. Results

## 3.1. Group and manipulation check data

The groups did not significantly differ on any baseline measure (table 1). On their Western-style dietary intake score, both groups fell around 0.7 s.d. below the normative mean for students ($M = 61.5$, s.d. = 13.5; $n = 2977$), indicating a generally healthier diet than average. Table 2 presents food intake-related data on Day 1 and 8, and during the intervention (Days 2–7). On Days 1 and 8, breakfast energy intake was higher for the WS-diet group ($p < 0.001$), but did not differ by days, group or group by days. Breakfast intake volume did not differ by days, group or group by days. Blood glucose significantly increased across breakfast ($p < 0.001$). Blood glucose change scores did not differ by days, group or group by days. Hunger, fullness and thirst ratings altered across breakfast (all $p$'s < 0.001), but change across breakfast scores did not differ by days, group or group by days. There was no effect of group, days or group by days, on the breakfast evaluations, indicating that all participants found it generally pleasant and filling.

During the intervention period, the WS-diet group reported eating most of the target food (fast-food $M = 92.5\%$ [95% CI, 88.3–96.7]; waffles $M = 88.0\%$ [95% CI, 83.9–92.0]), the majority returned fast-food receipts (60.0% (both receipts); 75% (at least one receipt)) and photographs of themselves eating the waffles (89.0% (all occasions)). As expected, (table 2) the WS-diet group's food diaries revealed intake differences when compared to controls, notably with higher saturated fat ($p < 0.001$) and sugar intake ($p = 0.003$), and greater overall energy intake ($p = 0.003$).

## 3.2. Appetitive control: the wanting and liking test

Change in wanting, liking and want more ratings across state (i.e. across breakfast) for the palatable breakfast foods, were analysed using a three-way mixed-design ANOVA, with Rating type (wanting versus liking versus want more) and Day (Day 1 versus Day 8) as within-factors and Group (WS-diet versus control) as the between factor (figure 1). The ANOVA revealed a main effect of Rating type, $F_{2,206} = 78.01$, MSE = 259.95, $p < 0.001$, $d = 1.74$, with liking ratings changing significantly less across state than wanting and want more ratings, with this clearly evident for each group on each day (see figure 1a–d). Crucially, we observed just one further effect, the predicted interaction of Rating type, Day and Group, $F_{2,206} = 3.56$, MSE = 154.97, $p = 0.03$, $d = 0.37$.

To determine the origin of this interaction effect we conducted a further four analyses. First, we repeated the three-way mixed-design ANOVA, but now contrasting: (a) wanting and liking ratings;

**Table 2.** Descriptive statistics ($M$ and 95% CI) for intake-related variables.

| day and variable | WS-diet group | control group |
| --- | --- | --- |
| **Day 1 (laboratory breakfast)** | | |
| energy intake (KJ) | 3495.5 (3381.2–3609.8) | 2485.6 (2407.8–2563.4)[b] |
| volume intake ($cm^3$) | 557.7 (539.7–575.6) | 549.9 (534.0–565.7) |
| $\Delta$ blood glucose (mmol l$^{-1}$)[a] | 1.9 (1.6–2.1) | 1.7 (1.4–1.9) |
| $\Delta$ hunger[a] | −52.5 (−44.7–60.3) | −55.6 (−47.7–63.5) |
| $\Delta$ fullness[a] | 56.7 (46.5–67.0) | 60.4 (52.5–68.2) |
| $\Delta$ thirst[a] | −6.5 (3–16.1) | −16.3 (-7.5–25.2) |
| like breakfast | 64.9 (56.4–73.4) | 65.6 (57.6–73.6) |
| how filling | 95.9 (89.0–102.8) | 100.0 (95.6–104.4) |
| **Day 2–7 (diary estimated average daily intake)** | | |
| energy intake (KJ) | 10127.4 (9338.9–10916.3) | 8548.8 (7745.1–9292.5)[b] |
| % protein (total) | 23.3 (22.0–24.6) | 26.5 (24.6–28.3)[b] |
| % fat (total) | 23.3 (22.3–24.4) | 19.5 (17.9–21.0)[b] |
| % saturated fat | 9.6 (9.1–10.1) | 6.8 (6.1–7.6)[b] |
| % carbohydrate (total) | 53.4 (51.7–55.1) | 54.1 (51.7–56.4) |
| % sugar | 25.0 (23.3–26.6) | 20.4 (17.9–22.8)[b] |
| **Day 8 (laboratory breakfast)** | | |
| energy intake (KJ) | 3437.6 (3321.0–3554.1) | 2507.4 (2427.7–2587.0)[b] |
| volume intake ($cm^3$) | 548.5 (530.3–566.7) | 553.5 (537.5–569.4) |
| $\Delta$ blood glucose (mmol l$^{-1}$)[a] | 1.9 (1.7–2.1) | 1.9 (1.7–2.2) |
| $\Delta$ hunger[a] | −47.2 (−36.5–57.8) | −58.8 (−51.4–66.1) |
| $\Delta$ fullness[a] | 62.5 (52.5–72.6) | 70.2 (63.6–76.8) |
| $\Delta$ thirst[a] | −13.2 (−4.5–21.9) | −13.9 (−6.9–20.9) |
| like breakfast | 67.1 (58.8–75.3) | 66.3 (59.1–73.5) |
| how filling | 101.1 (95.3–106.8) | 100.1 (96.1–104.2) |

[a]Post minus pre.
[b]Significantly different between groups, $p < 0.05$.

(b) want more and liking ratings; and (c) wanting and want more ratings. Consistent with expectations, the only occasion where the Rating type by Day by Group interaction was now significant (all other $p$'s > 0.13), was when wanting and liking ratings were contrasted, $F_{1,103} = 5.81$, MSE = 189.38, $p = 0.018$, $d = 0.47$. These data are illustrated in figure 1e,f. Second, we tested within the WS-diet group whether the pre- to post-breakfast change in wanting and liking differed between Days 1 and 8 (i.e. data in figure 1f). This effect was significant, $F_{1,53} = 6.59$, MSE = 183.18, $p = 0.013$, $d = 0.71$, indicating that the difference between wanting and liking ratings were significantly reduced across breakfast on Day 8 relative to Day 1, in the WS-diet group. Importantly, this pattern of findings mirrors that observed cross-sectionally, between participants who habitually eat a WS-diet versus those who consume a healthier diet [15]. Third, we contrasted just the Day 8 data between groups, for the difference between wanting and liking across breakfast (i.e. right-hand part of figure 1e versus right-hand part of figure 1f). This too was significant, $F_{1,103} = 8.11$, MSE = 216.11, $p = 0.005$, $d = 0.56$, indicating that performance on Day 8, following the diet intervention, impacted the WS-diet group by reducing the difference between wanting and liking ratings across breakfast. Fourth, we examined whether the pre-breakfast only data differed between groups, using all three scales, as there appears to be some lowering of all initial ratings in the WS-diet group. This analysis found no interaction effects involving group (all $p$'s > 0.13), only a main effect of this variable—group, $F_{1,103} = 5.43$, MSE = 2489.02, $p = 0.022$, $d = 0.46$—indicating a general propensity for lower wanting, liking and want more ratings in the WS-diet group on both days before breakfast. Finally, as energy intake differed across breakfast, we repeated the main analyses using it as a covariate, finding that it enlarged key effect sizes.

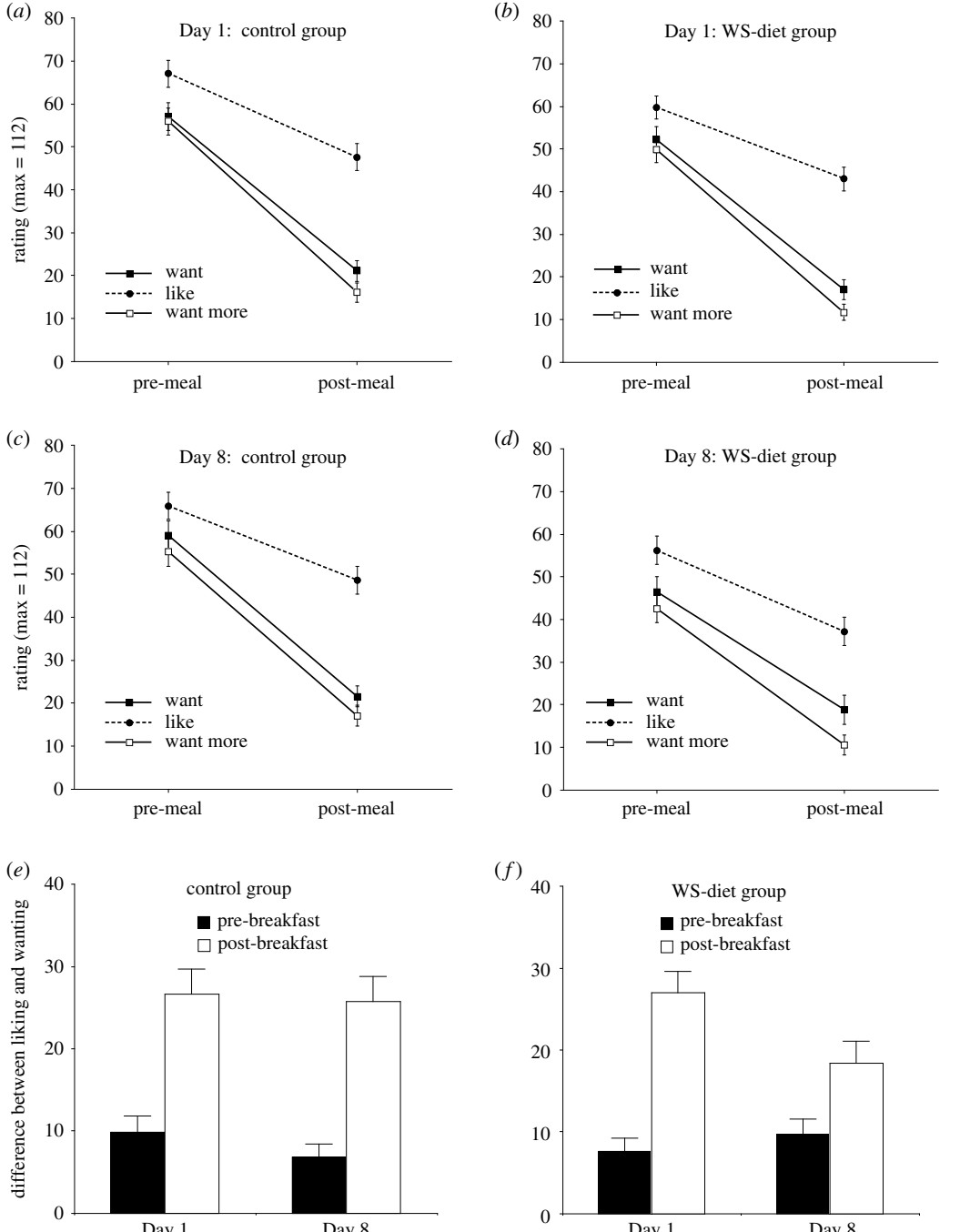

**Figure 1.** Mean (and s.e.) ratings for the wanting and liking test, on Day 1 (*a,b*) and Day 8 (*c,d*) before and after the meal (breakfast) for each group and the key difference between liking and wanting ratings, before and after breakfast, on Day 1 and 8, for the controls (*e*) and for the WS-diet group (*f*). In (*e,f*), it can be seen that the difference between changes in wanting and liking across breakfast are significantly smaller in the WS-diet group following the intervention (Day 8), while remaining unchanged in controls.

## 3.3. Neuropsychological testing

HVLT learning and retention scores are presented in table 3. Bootstrapped univariate ANCOVA with HVLT delayed retention score on Day 8 as the dependent variable, group as the independent variable and Day 1 score as the covariate, revealed a significant main effect of group, $F_{1,102} = 4.72$, MSE = 138.30, $p = 0.032$, $d = 0.43$. The WS-diet group had a significantly lower retention score on Day 8 ($M = 92.0\%$ [95% CI, 88.5–95.4]) than controls ($M = 97.0\%$ [95% CI, 94.2–99.9]). The same ANCOVA design revealed no difference in digit span score by group on Day 8 (WS-diet $M = 14.3$ [95% CI, 13.4–15.3]; controls $M = 14.7$ [95% CI, 13.4–16.1]).

**Table 3.** HVLT performance across the experiment.

| day and variable | WS-diet group | control group |
|---|---|---|
| Day 1 | | |
| trial 1 | 6.4 (5.9–7.0) | 6.8 (6.3–7.2) |
| trial 2 | 9.2 (8.7–9.7) | 9.2 (8.7–9.7) |
| trial 3 | 10.2 (9.6–10.7) | 10.2 (9.7–10.6) |
| delayed trial | 9.6 (9.0–10.2) | 9.6 (9.2–10.1) |
| %delayed retention | 93.1 (90.5–95.7) | 93.8 (91.1–96.5) |
| Day 8 | | |
| trial 1 | 6.3 (5.8–6.8) | 6.5 (6.1–6.9) |
| trial 2 | 8.9 (8.4–9.5) | 9.5 (9.0–10.0) |
| trial 3 | 9.9 (9.4–10.5) | 10.3 (9.8–10.7) |
| delayed trial | 9.3 (8.7–9.9) | 10.0 (9.5–10.4) |
| %delayed retention | 92.0 (88.5–95.4) | 97.0 (94.2–99.9) |
| Day 29 | | |
| trial 1 | 6.7 (6.1–7.2) | 6.7 (6.2–7.1) |
| trial 2 | 9.4 (8.9–10.0) | 9.2 (8.7–9.7) |
| trial 3 | 10.5 (10.0–11.0) | 10.1 (9.6–10.7) |
| delayed trial | 9.8 (9.2–10.4) | 9.7 (9.1–10.3) |
| %delayed retention | 92.4 (89.4–95.1) | 94.2 (91.4–96.5) |

We then tested whether Day 29 HVLT retention score differed by group, using the same ANCOVA design, but there was now no group effect (WS-diet $M = 92.4\%$ [95% CI, 89.4–95.1]; control $M = 94.2\%$ [95% CI, 91.4–96.5]). The same analysis approach revealed no difference by group in digit span score on Day 29 (WS-diet $M = 14.5$ [95% CI, 13.5–15.6]; control $M = 14.9$ [95% CI, 13.6–16.2]).

## 3.4. Relationship between HDLM and appetitive control measure: wanting and liking

We calculated a single score for each participant reflecting the key outcome from the wanting and liking test (i.e. the data displayed in figure 1e,f; the Rating type [wanting versus liking] × Day [1 versus 8] × group effect)—the appetitive control score. This score was the difference between liking and wanting ratings after breakfast, minus that before breakfast, with this value being computed for Day 1 and Day 8. The Day 1 score was then subtracted from the Day 8 score, yielding a mean value of 9.5 [95% CI, 2.1–16.8] in the WS-diet group and −3.5 [95% CI, −11.4–4.4] in controls.

Spearman's $\rho$ (due to non-normality), revealed a significant relationship between the appetitive control score (i.e. the key outcome on the wanting and liking test), and change in HVLT delayed retention score by Day. A larger reduction in HVLT score across the study was associated with a reduced effect of state on wanting and liking ratings, $\rho$ [105] = 0.25, $p = 0.01$, $d = 0.52$. As this relationship should only be evident in the WS-diet group, we tested for it there (figure 2), where it was significant, $\rho$ [54] = 0.46, $p < 0.001$, $d = 1.01$. There was no relationship in controls ($\rho = 0.01$; figure 2). The difference in correlations between groups was reliable, $Z = 2.35$, $p < 0.02$, $d = 0.47$.

# 4. Discussion

One week's exposure to a WS-diet caused a measurable weakening of appetitive control, as measured by the two key ratings on the wanting and liking test. Prior to the intervention, participants viewed palatable breakfast foods and judged how much they wanted to eat them, and then how much they liked their actual taste. This test was repeated after participants had eaten to satiety. Across these pre- and post-meal tests, wanting ratings declined far more than ratings of taste liking. This manifestation of appetitive control—that is the expectation that food is less desirable than it *actually* tastes—changed

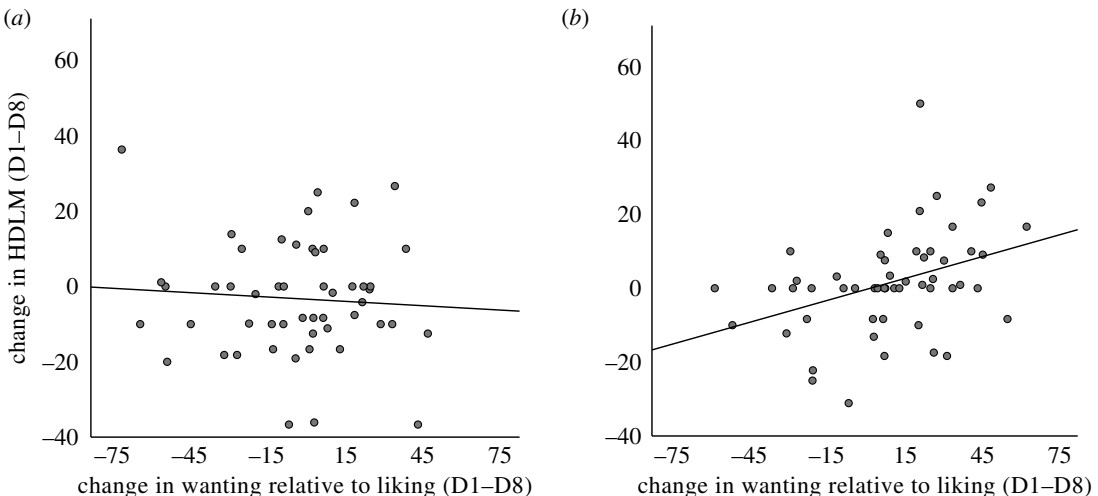

**Figure 2.** Scatter plots (with linear fitted slope) of change in performance on the measure of hippocampal-dependent learning and memory (HDLM; Hopkins verbal learning test), between Day 1 and 8, and the change in wanting relative to liking across breakfast, between Day 1 and 8, for the control (a) and WS-diet (b) groups.

in participants following the Western-style dietary intervention. When sated, the WS-diet group reported an equivalent decline in wanting and taste liking. This finding parallels cross-sectional data, with performance on the wanting and liking test in the WS-diet group here coming to resemble that of the habitual WS-diet consumers in Attuquayefio et al.'s [15] study. We also found that a WS-diet induced a decline in HDLM performance in the WS-diet group. This replicates a previous demonstration of this effect [7] and parallels findings in the extensive animal literature (e.g. [1–6]). The observed decline in HDLM strongly correlated with the change in appetitive control measured by the wanting and liking test, suggesting a probable common hippocampal basis for this effect. We also examined the reversibility of the diet-induced HDLM decline, finding that the group difference was no longer significant at three weeks post-intervention. However, it is unclear whether this reflects a deterioration in performance in the controls or recovery in the WS-diet group. While this study cannot firmly conclude which possibility may be correct, we note that reversibility is observed in animal studies that use short-term Western-style dietary exposures [36].

Before turning to the interpretation of these data, two issues require consideration. The first concerns the reliability and validity of dietary manipulations. Participants often under-report food intakes (e.g. [48]), their compliance with dietary interventions can be poor (e.g. [49]) and there is often a high drop-out rate due to costs, the reduced palatability of the diet, and the lengthy (two to three months) nature of the intervention (e.g. [50]). These oft-cited concerns have limited relevance to the current study. Our study was brief, it did not ask participants to restrict their food intake or make more healthful choices, participants were under no obligation to follow the experimental instructions (i.e. they could drop out and still receive payment/course credit) and the dietary changes were cost neutral. The majority of participants returned receipts indicating they had purchased the fast-food meals as well as providing photographs of themselves eating the waffles. These measures were further confirmed by diet diaries, which indicated increased energy intake, sugar and saturated fat consumption, as would be expected if the intervention had been enacted. Such dietary self-reports can be quite accurate, especially in normal-weight participants (e.g. [51,52]). Together, we suggest that participants in the WS-diet group generally complied with the intervention and had little reason not to do so.

A second issue concerns the wanting and liking test. This test was conducted before breakfast on a set of palatable test foods, and then again after breakfast. However, the breakfast that each group received differed in energy and macronutrient content. Thus, it is conceivable that this might account for the group difference observed on the wanting and liking test, in conjunction with putative reductions in interoceptive sensitivity in the WS-diet group. We suggest this is unlikely. First, the group difference was only evident on Day 8, yet breakfast remained unaltered for both groups. Second, there were no differences in volume or palatability between the breakfasts on either Day 1 or 8. Third, and as noted in the results, controlling for energy intake at breakfast serves to increase the size of the group difference not diminish it. Fourth, again as noted in the Results, hunger and fullness ratings did not

significantly differ by group, time of testing or by their interaction. While it would have been ideal to use the same breakfast, this had to be balanced against generating a diet change in the WS-diet group while minimizing participant burden.

A final point here concerns the validity of the wanting and liking test. Our working assumption was that changes in performance on this test would translate into changes in eating behaviour. Thus the change in wanting relative to liking in the WS-diet group should reflect a greater likelihood of eating more when sated. As the wanting and liking test is relatively new, its capacity to predict over-eating when sated has not been directly tested. However, its ecological validity is suggested by the intersection of two observations. First, pre- and post-intervention performance on the wanting and liking test in the WS-diet group mirrors the cross-sectional difference seen between infrequent and frequent consumers of Western-style foods [15]. Second, frequent consumers of Western-style foods tend to eat more than infrequent consumers, and especially of palatable snack foods (e.g. [17,53]). Together, this suggests that over-eating occurs in consumers of a WS-diet, and that such consumers also demonstrate smaller reduction in wanting relative to liking across a meal.

In the Introduction, we outlined why state-dependent modulation was important, and how it might be achieved in two different models. One model was premised around a dissociation between brain areas underpinning anticipatory (i.e. wanting) and consummatory (i.e. liking) reward, with activation of wanting substrates by excitatory cues driving goal-seeking behaviour (e.g. [20–22]). The other model proposed that excitatory cues activate associative networks connected with obtaining that food, including memories of its affective and sensory properties, and behavioural outcomes (e.g. [3,12,13]). We also noted that inhibitory hippocampal modulation of appetite in animals could occur independent of state [24], perhaps paralleling memory-based inhibition effects in people [11]. Returning to the accounts outlined in the Introduction, the incentive salience model was suggested to occur via internal state acting to dampen brain reward circuit activity. In the other account, modulation occurred via state-dependent inhibition of associative networks connected with obtaining a food (e.g. [3,12,13]). While the data from our study suggests that the hippocampus is involved in modulation, we did not set out to contrast these models. Nonetheless, one finding from the wanting and liking test is of interest in this regard. If state exerts a dampening effect on brain reward circuits, it would presumably do so on both wanting *and* liking—at least there seems to be nothing in the incentive salience account that requires or specifies differential dampening. By contrast, if state acts to inhibit retrieval of food-related associative networks, the effect would be limited to wanting. As the wanting and liking data here, and from previous uses of this test [15,45,46], consistently point to a greater reduction in wanting relative to liking after a meal, this outcome seems more consistent with an associative inhibition model.

In animals, there are extensive neural connections between the hippocampus and cortical, olfactory, hypothalamic, mid- and hind-brain, and subcortical structures, including the nucleus accumbens ([23], fig. 1). The human hippocampus has a similarly extensive network [54], including connections to the nucleus accumbens [55]. In theory, this extensive network could support either model, in allowing the hippocampus to modulate brain reward areas (e.g. nucleus accumbens), in having the relevant in-flow of contextual information (i.e. internal bodily state; e.g. [29]) and its involvement in memory and inhibitory processing (e.g. [25]). More broadly, not only are striatal and hippocampal circuits connected, but both areas extensively link to the frontal cortex in animals and humans [25,56]. In animals, these three brain areas interact to control behaviour, such as when there is an interplay between striatal-based procedural memory (e.g. moving to a goal) and hippocampal-based declarative memory (e.g. cue-based navigation), with prefrontal cortical control (e.g. [57,58]). In humans, and from the perspective of appetitive behaviour, a different form of interaction may be hypothesized, with the hippocampus exerting modulatory control over striatal reward/goal-movement circuits, with frontal oversight. If hippocampal modulatory control were to wane, then regulatory oversight may then depend more on frontal circuits, implying a greater dependence on explicit self-control to regulate appetite.

Another way of conceptualizing the findings from the wanting and liking test, is that wanting ratings reflect the reward value of the goal object, its desirability (see [59]). The self-control literature has generally been less interested in whether the reward value of a goal object can be a significant factor in self-control failure. Rather, the emphasis has more been upon the capacity of a person to inhibit their response to that goal object. As noted above, the neural basis of self-control focuses heavily on frontal circuits involved in response inhibition (e.g. [60,61]). If goal object value can become less sensitive to internal state as our results imply, then this would presumably increase the likelihood of a self-control failure, by making the object, on average, more appealing. This, in turn, would then

require more cognitive effort to inhibit a response. As hippocampal-related roles in reward value are generally not to the fore when considering self-control, the findings here and from animal studies (e.g. [3,12,13,62,63]) would suggest that they may be worthy of further consideration.

Consistent with Attuquayefio et al. [7] we too observed a decline in HDLM performance, as measured on both occasions by delayed retention on the HVLT, following a Western-style dietary intervention. The performance decrement in both studies was of similar magnitude ($d = 0.46$ in [7]; and $d = 0.43$ here), representing a small to intermediate effect size. While it may seem surprising that a brief Western-style dietary intervention can affect HDLM performance, it should be noted that the foundations for this claim are quite extensive, as we detailed above and in the Introduction.

In our previous study [7], we observed that increases in blood glucose were greater across breakfast in the WS-diet group relative to controls, but this was not observed here. While this may suggest that other components of the dietary intervention (e.g. reduction in protein) might be responsible for the effects on HDLM, the extensive animal literature suggests that it is changes in saturated fat and added sugar that are important. Moreover, the animal literature has now started to establish why saturated fats and added sugar—and not other macronutrients—should have this effect [4,14,64]. Western-style meals generate repeated mild inflammatory responses, caused in part by their effect on blood glucose and oxidative stress, among other changes (e.g. [14]). Together these can selectively weaken the integrity of the blood-brain barrier in the region of the hippocampus—although this weakening may require longer time periods than here (e.g. [64,65]). A local hippocampal-based inflammatory response may then ensue, adversely affecting neural plasticity and neurogenesis (e.g. [4]).

The reduction in HDLM in the WS-diet group was strongly associated with change in performance on the wanting and liking test (i.e. the effect illustrated in figure 1f). As described above, we interpreted this correlation to mean that both of these tests reflected hippocampal function, manifesting via state-based inhibition. However, this is not the only way this finding could be interpreted. WS-diet could impair two independent systems to a similar degree, if, for example, there was a common underlying vulnerability factor. This possibility is lent some weight by the finding noted earlier that hippocampal inhibition of appetite can occur independent of state [24]. As state provides an important link tying together the HDLM and wanting and liking findings, one implication of Hannapel et al. [24] is that state-based changes in wanting and liking in the WS-diet group could be a separate phenomenon. As the WS-diet manipulation involved increased consumption of palatable foods, it is plausible that this could alter brain reward processes thus affecting wanting and liking judgements. However, while little is known about the neural basis of state-dependent changes in wanting and liking and thus whether they are hippocampally mediated (or not), two considerations favour a link between impaired HDLM and impaired performance on the wanting and liking test. The first is that presuming they have a common cause is more parsimonious than assuming two independent causes. The second concerns the relationship between memory, interoception and the hippocampus. It has been suggested that generating interoceptive states requires comparison of an expected state (i.e. one based on memory) with incoming interoceptive information [66]. On this basis, impairment of memory implies impairment of interoception, making a link between state-dependent effects (i.e. for wanting and liking) and hippocampal function. As this is an emerging area, and with much still to be learned about how these processes interrelate, our conclusions are of course tentative.

A number of authors have suggested that the hippocampus is especially vulnerable to environmental insults. Factors identified to impair hippocampal function include, type II diabetes (e.g. [67]), insomnia (e.g. [68]), stress (e.g. [69]), exposure to environmental toxins (e.g. [70]) and depression (e.g. [71]). All of these factors are common in Western societies and may combine with a WS-diet to cause acute, and in the longer-term, cumulative damage to the hippocampus. It may be for this reason that a WS-diet [72]—along with the other factors listed here—are known correlates of Alzheimer's disease [73,74].

In conclusion, a large animal literature demonstrates that a WS-diet adversely affects the hippocampus. The current study suggests something similar occurs in humans, in that one week's exposure to a WS-diet causes a reduction in HDLM performance, in addition to alterations in appetitive control, as measured by the wanting and liking test. The magnitude of these changes in HDLM and appetitive control were strongly correlated, implying a probable common basis for these effects in the hippocampus, and thus a role for the hippocampus in appetitive control. More broadly, this experiment, alongside those from the other animal and human studies cited here, suggests that a WS-diet causes neurocognitive impairments following short-term exposure.

Ethics. The procedure was approved by the Macquarie University Human Research Ethics Committee (approval 5201820352408). Written consent was provided by each participant.

Data accessibility. Data have been uploaded as electronic supplementary material, Excel file.

Authors' contributions. All authors contributed to the study design, interpretation of the findings and provided critical feedback during manuscript preparation. Data collection was overseen by R.J.S., H.M.F. and D.G., R.J.S. analysed the data and R.J.S. and H.M.F. drafted the initial manuscript. All authors approved the final manuscript.

Competing interests. The authors have no competing interests.

Funding. This work was supported by a grant from the Australian Research Council, DP150100105.

Acknowledgements. We thank Andrea Zuniga, Lina Teichmann, Zoe Taylor, Selene Petit and Kate Hardwick for assistance with this study.

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
