## [Reviewer comments · Royal Society Open Science]

Review History

RSOS-191338.R0 (Original submission)

Review form: Reviewer 1

Is the manuscript scientifically sound in its present form?

No

Are the interpretations and conclusions justified by the results?

No

Is the language acceptable?

Yes

Do you have any ethical concerns with this paper?

No

Have you any concerns about statistical analyses in this paper?

No

Recommendation?

Major revision is needed (please make suggestions in comments)

Comments to the Author(s)

Stevenson et al examine the effect of one week Western style diet feeding on hippocampal dependent learning and memory (HDLM), and appetitive control, before and after the feeding intervention. They report that HDLM and appetitive control declined following the Western diet and suggest that Western diet has adverse neurocognitive effects in humans as observed in preclinical models.

It would be useful to provide the reader with a table showing the HVLT recall on learning trials, and the delayed recall (means \pm SD), as well as the %retention. It would also be useful to provide the norms for this test and address whether the effects of diet are statistically significant but fall within expected normal range for function.

Table 1 provides the Western Diet score for the two study groups which are 50.7 and 52.9. What is the average score for this test and how much below the average are these scores?

From the data presented in Figure 1 it appears that wanting of test foods in those consuming western diet was less before the meal, yet declined to the same rating after the meal. Thus it is not clear that this data supports subjects wanting more after finishing the meal? Please explain.

Review form: Reviewer 2 (Michael Kendig)

Is the manuscript scientifically sound in its present form?

Yes

Are the interpretations and conclusions justified by the results?

Yes

Is the language acceptable?

Yes

Do you have any ethical concerns with this paper?

No

Have you any concerns about statistical analyses in this paper?

No

Recommendation?

Accept with minor revision (please list in comments)

Comments to the Author(s)

Stevenson and colleagues examine the effects of a 1-week western-style diet (WD) intervention on hippocampal-dependent learning and memory (HDLM) and measures of 'wanting' and 'liking' of various foods. The WD intervention significantly reduces performance on a measure of HDLM but not on a digit-span task, replicating prior work by this group. To measure appetitive control participants rated how much they liked, wanted and wanted more of various foods before and after a breakfast. The authors compare pre- and post-breakfast change scores before and after the diet intervention, and claim that shifts in wanting vs. liking ratings on the day 8 test is evidence of weakened appetitive control produced by exposure to the WD.

The experiment is well-designed and the manuscript is clearly written. Results have interesting implications for the intersection of hippocampal-dependent learning and memory processes with acute appraisal to food reward value.

Major comments

- 1) The conclusion that HDLM performance recovered at follow up does not seem supported by the data. The absence of a Group effect on day 29 appears due to poorer performance in the control group (which decreased from 97 to 94.2%) rather than an increase in the WD group, whose performance was stable (92 to 92.4%). This conclusion should be revised.
- 2) Could the differing energy content and macronutrient composition of the control and WD lab breakfasts confound interpretation of the pre-post changes on want/like measures? The WD breakfast produces smaller changes in hunger, fullness and thirst on day 1 and day 8, which could feasibly alter post-breakfast ratings. I wonder whether the proposed 'weakening of appetitive control' is not some artefact of meal composition. This could be discussed.
- 3) The results of the wanting and liking test are somewhat difficult to interpret. The 3-way interaction found for the wanting/liking test, and subsequent interaction with only wanting vs. liking, gives good evidence for WD effects. However these analyses are examining changes in change scores, and even the 'follow-up' interaction appears to have 4 factors: wanting vs. liking, pre vs. post, WD vs. control, day 1 vs. day 8. As such it still isn't clear what drives the interaction, and I think further analyses would help bolster the authors' case. Some suggestions (but others may be more appropriate):
 - a. Within the WD group, does the pre-post change in wanting vs liking change from day 1 to day 8? I.e., directly test the hypothesis that the drop in liking becomes more pronounced only in WD participants.
 - b. Considering only day 8 data, is the group x scale (want vs. like) interaction significant?
 - c. Comparing pre-meal measures between groups on day 1 vs. 8. The WD group appears to make lower pre-meal ratings on all 3 subscales on both days, particularly on day 8 (although here there might be reason to expect group differences). Was this significant?
 - d. Showing the significant difference graphically would greatly help the reader interpret the figure.
- 4) Digit span: performance seems uncannily high on this task (13-15 numbers recalled) – do the means refer to the number of digits recalled?
- 5) Lines 341-348: this analysis looks interesting, and the correlations seem robust, but which change score is being evaluated – wanting, liking, or wanting more? Only one rho value is provided.
- 6) In several instances non-significant results are dismissed without further comment. Such results still hold great value and may be of interest to those in the field (e.g. there is a growing field examining diet effects on mood). Adding even descriptive data (in supplementary form) would permit their inclusion in meta-analyses and would be a valuable asset.

Minor comments

Line 74: suggest changing 'a week' to 'four days'.

Line 112-115: this sentence was difficult to follow – suggest splitting and/or rephrasing.

Line 183: please define what is meant by 'below average' – did this refer to the average (mean/median?) of the present sample, or to reference data from Francis & Stevenson, 2013?

Line 201: what instructions were given to the WS-diet group regarding their other meals?

Line 212: Is it true that the control breakfast was 58% protein? Could protein and carbohydrate proportions have been swapped mistakenly?

Line 215: Please define 'Mini-toast'. 'Froot', not Fruit loops (regrettably) is the spelling

Line 248: word missing, 'how much more'

Lines 321-323: Please add the results of these statistical tests

Line 443: typo, 'Atuquayefio'

Line 454: it could be added that altered BBB permeability tends to be observed only after long diet exposures (i.e., after cognitive impairment).

Review form: Reviewer 3 (Marise Parent)

Is the manuscript scientifically sound in its present form?

Yes

Are the interpretations and conclusions justified by the results?

No

Is the language acceptable?

Yes

Do you have any ethical concerns with this paper?

No

Have you any concerns about statistical analyses in this paper?

No

Recommendation?

Accept with minor revision (please list in comments)

Comments to the Author(s)

See attached file (Appendix A).

Decision letter (RSOS-191338.R0)

04-Nov-2019

Dear Professor Stevenson,

The editors assigned to your paper ("Hippocampal-dependent appetitive control is impaired by experimental exposure to a Western-style diet") have now received comments from reviewers. We would like you to revise your paper in accordance with the referee and Associate Editor suggestions which can be found below (not including confidential reports to the Editor). Please note this decision does not guarantee eventual acceptance.

Please submit a copy of your revised paper before 27-Nov-2019. Please note that the revision deadline will expire at 00.00am on this date. If we do not hear from you within this time then it will be assumed that the paper has been withdrawn. In exceptional circumstances, extensions may be possible if agreed with the Editorial Office in advance. We do not allow multiple rounds of revision so we urge you to make every effort to fully address all of the comments at this stage. If deemed necessary by the Editors, your manuscript will be sent back to one or more of the original reviewers for assessment. If the original reviewers are not available, we may invite new reviewers.

When submitting your revised manuscript, you must respond to the comments made by the

referees and upload a file "Response to Referees" in "Section 6 - File Upload". Please use this to document how you have responded to the comments, and the adjustments you have made. In order to expedite the processing of the revised manuscript, please be as specific as possible in your response.

- Data accessibility

If you wish to submit your supporting data or code to Dryad (<http://datadryad.org/>), or modify your current submission to dryad, please use the following link:
<http://datadryad.org/submit?journalID=RSOS&manu=RSOS-191338>

- Competing interests

- Authors' contributions

- Acknowledgements

- Funding statement

Kind regards,
Andrew Dunn
Senior Publishing Editor
Royal Society Open Science
openscience@royalsociety.org

on behalf of Prof Essi Viding (Subject Editor)
openscience@royalsociety.org

Associate Editor's comments:

Thank you for submitting this paper to Royal Society Open Science for consideration. We've received commentary from 3 reviewers, and the general view is that the manuscript is heading in the right direction for publication, but that you've a number of revisions required before it may be ready for acceptance. Please ensure that you fully respond to the queries and concerns of the reviewers, and make sure you include not only a tracked changes version of the paper but also a full point-by-point response with your revision to help the Editors and reviewers assess how well you've addressed their concerns. Good luck!

Associate Editor: 2
Comments to the Author:
(There are no comments.)

Comments to Author:

Reviewers' Comments to Author:
Reviewer: 1

Comments to the Author(s)

Stevenson et al examine the effect of one week Western style diet feeding on hippocampal dependent learning and memory (HDLM), and appetitive control, before and after the feeding intervention. They report that HDLM and appetitive control declined following the Western diet and suggest that Western diet has adverse neurocognitive effects in humans as observed in preclinical models.

It would be useful to provide the reader with a table showing the HVLT recall on learning trials, and the delayed recall (means \pm SD), as well as the %retention. It would also be useful to provide the norms for this test and address whether the effects of diet are statistically significant but fall within expected normal range for function.

Table 1 provides the Western Diet score for the two study groups which are 50.7 and 52.9. What is the average score for this test and how much below the average are these scores?

From the data presented in Figure 1 it appears that wanting of test foods in those consuming western diet was less before the meal, yet declined to the same rating after the meal. Thus it is not clear that this data supports subjects wanting more after finishing the meal? Please explain.

Reviewer: 2

Comments to the Author(s)

Stevenson and colleagues examine the effects of a 1-week western-style diet (WD) intervention on hippocampal-dependent learning and memory (HDLM) and measures of 'wanting' and 'liking'

of various foods. The WD intervention significantly reduces performance on a measure of HDLM but not on a digit-span task, replicating prior work by this group. To measure appetitive control participants rated how much they liked, wanted and wanted more of various foods before and after a breakfast. The authors compare pre- and post-breakfast change scores before and after the diet intervention, and claim that shifts in wanting vs. liking ratings on the day 8 test is evidence of weakened appetitive control produced by exposure to the WD.

The experiment is well-designed and the manuscript is clearly written. Results have interesting implications for the intersection of hippocampal-dependent learning and memory processes with acute appraisal to food reward value.

Major comments

- 1) The conclusion that HDLM performance recovered at follow up does not seem supported by the data. The absence of a Group effect on day 29 appears due to poorer performance in the control group (which decreased from 97 to 94.2%) rather than an increase in the WD group, whose performance was stable (92 to 92.4%). This conclusion should be revised.
- 2) Could the differing energy content and macronutrient composition of the control and WD lab breakfasts confound interpretation of the pre-post changes on want/like measures? The WD breakfast produces smaller changes in hunger, fullness and thirst on day 1 and day 8, which could feasibly alter post-breakfast ratings. I wonder whether the proposed 'weakening of appetitive control' is not some artefact of meal composition. This could be discussed.
- 3) The results of the wanting and liking test are somewhat difficult to interpret. The 3-way interaction found for the wanting/liking test, and subsequent interaction with only wanting vs. liking, gives good evidence for WD effects. However these analyses are examining changes in change scores, and even the 'follow-up' interaction appears to have 4 factors: wanting vs. liking, pre vs. post, WD vs. control, day 1 vs. day 8. As such it still isn't clear what drives the interaction, and I think further analyses would help bolster the authors' case. Some suggestions (but others may be more appropriate):
 - a. Within the WD group, does the pre-post change in wanting vs liking change from day 1 to day 8? I.e., directly test the hypothesis that the drop in liking becomes more pronounced only in WD participants.
 - b. Considering only day 8 data, is the group x scale (want vs. like) interaction significant?
 - c. Comparing pre-meal measures between groups on day 1 vs. 8. The WD group appears to make lower pre-meal ratings on all 3 subscales on both days, particularly on day 8 (although here there might be reason to expect group differences). Was this significant?
 - d. Showing the significant difference graphically would greatly help the reader interpret the figure.
- 4) Digit span: performance seems uncannily high on this task (13-15 numbers recalled) – do the means refer to the number of digits recalled?
- 5) Lines 341-348: this analysis looks interesting, and the correlations seem robust, but which change score is being evaluated – wanting, liking, or wanting more? Only one rho value is provided.
- 6) In several instances non-significant results are dismissed without further comment. Such results still hold great value and may be of interest to those in the field (e.g. there is a growing field examining diet effects on mood). Adding even descriptive data (in supplementary form) would permit their inclusion in meta-analyses and would be a valuable asset.

Minor comments

Line 74: suggest changing 'a week' to 'four days'.

Line 112-115: this sentence was difficult to follow – suggest splitting and/or rephrasing.

Line 183: please define what is meant by 'below average' – did this refer to the average (mean/median?) of the present sample, or to reference data from Francis & Stevenson, 2013?

Line 201: what instructions were given to the WS-diet group regarding their other meals?

Line 212: Is it true that the control breakfast was 58% protein? Could protein and carbohydrate proportions have been swapped mistakenly?

Line 215: Please define 'Mini-toast'. 'Froot', not Fruit loops (regrettably) is the spelling

Line 248: word missing, 'how much more'

Lines 321-323: Please add the results of these statistical tests

Line 443: typo, 'Atuquayefio'

Line 454: it could be added that altered BBB permeability tends to be observed only after long diet exposures (i.e., after cognitive impairment).

Reviewer: 3

Comments to the Author(s)

See attached file.

Author's Response to Decision Letter for (RSOS-191338.R0)

See Appendix B.

RSOS-191338.R1 (Revision)

Review form: Reviewer 1

Is the manuscript scientifically sound in its present form?

Yes

Are the interpretations and conclusions justified by the results?

Yes

Is the language acceptable?

Yes

Do you have any ethical concerns with this paper?

No

Have you any concerns about statistical analyses in this paper?

No

Recommendation?

Accept as is

Comments to the Author(s)

Reviewer thanks the authors for consideration and response to prior critique. No further comments/questions.

Review form: Reviewer 2 (Michael Kendig)

Is the manuscript scientifically sound in its present form?

Yes

Are the interpretations and conclusions justified by the results?

Yes

Is the language acceptable?

Yes

Do you have any ethical concerns with this paper?

No

Have you any concerns about statistical analyses in this paper?

No

Recommendation?

Accept with minor revision (please list in comments)

Comments to the Author(s)

Symbols indicating the significant group differences on day 8 could be added to Table 3.
The figure legends on the bottom row of Fig 1 are overlapping.

Decision letter (RSOS-191338.R1)

02-Jan-2020

Dear Professor Stevenson:

On behalf of the Editors, I am pleased to inform you that your Manuscript RSOS-191338.R1 entitled "Hippocampal-dependent appetitive control is impaired by experimental exposure to a Western-style diet" has been accepted for publication in Royal Society Open Science subject to minor revision in accordance with the referee suggestions. Please find the referees' comments at the end of this email.

The reviewers and Subject Editor have recommended publication, but also suggest some minor revisions to your manuscript. Therefore, I invite you to respond to the comments and revise your manuscript.

- Ethics statement

- Data accessibility

If you wish to submit your supporting data or code to Dryad (<http://datadryad.org/>), or modify your current submission to dryad, please use the following link:
<http://datadryad.org/submit?journalID=RSOS&manu=RSOS-191338.R1>

- **Competing interests**

- **Authors' contributions**

- **Acknowledgements**

- **Funding statement**

Because the schedule for publication is very tight, it is a condition of publication that you submit the revised version of your manuscript before 11-Jan-2020. Please note that the revision deadline will expire at 00.00am on this date. If you do not think you will be able to meet this date please let me know immediately.

1) A text file of the manuscript (tex, txt, rtf, docx or doc), references, tables (including captions) and figure captions. Do not upload a PDF as your "Main Document".

- 2) A separate electronic file of each figure (EPS or print-quality PDF preferred (either format should be produced directly from original creation package), or original software format)
- 3) Included a 100 word media summary of your paper when requested at submission. Please ensure you have entered correct contact details (email, institution and telephone) in your user account
- 4) Included the raw data to support the claims made in your paper. You can either include your data as electronic supplementary material or upload to a repository and include the relevant doi within your manuscript
- 5) All supplementary materials accompanying an accepted article will be treated as in their final form. Note that the Royal Society will neither edit nor typeset supplementary material and it will be hosted as provided. Please ensure that the supplementary material includes the paper details where possible (authors, article title, journal name).

on behalf of Prof Essi Viding (Subject Editor)
openscience@royalsociety.org

Associate Editor Comments to Author:

Overall, the reviewers seem happy with your changes, but have a couple of minor final suggestions for the presentation of the paper.

Reviewer comments to Author:

Reviewer: 2

Comments to the Author(s)

Symbols indicating the significant group differences on day 8 could be added to Table 3. The figure legends on the bottom row of Fig 1 are overlapping.

Reviewer: 1

Comments to the Author(s)

Reviewer thanks the authors for consideration and response to prior critique. No further comments/questions.

Author's Response to Decision Letter for (RSOS-191338.R1)

See Appendix C.

Decision letter (RSOS-191338.R2)

09-Jan-2020

Dear Professor Stevenson,

It is a pleasure to accept your manuscript entitled "Hippocampal-dependent appetitive control is impaired by experimental exposure to a Western-style diet" in its current form for publication in Royal Society Open Science.

You can expect to receive a proof of your article in the near future. Please contact the editorial office (opencscience_proofs@royalsociety.org) and the production office (opencscience@royalsociety.org) to let us know if you are likely to be away from e-mail contact -- if you are going to be away, please nominate a co-author (if available) to manage the proofing process, and ensure they are copied into your email to the journal.

Kind regards,
Lianne Parkhouse
Editorial Coordinator
Royal Society Open Science
opencscience@royalsociety.org

on behalf of the Associate Editor and Professor Essi Viding (Subject Editor)
opencscience@royalsociety.org

Appendix A

This study investigated whether a high fat/high sugar (AKA Western; WS) diet impairs hippocampal-dependent learning and memory and appetitive control in humans as it does in rodents. Healthy lean adults were randomized to a 1-week WS diet or continued with their habitual diet. Hippocampal function was measured using the Hopkins Verbal Learning Test; HVLT) and appetitive control was assessed using ratings of liking and wanting of palatable snack foods under conditions of hunger and satiety. Compared to the control/habitual diet, the WS impaired HVLT performance and decreased the effects of satiety on wanting/liking, and the effects of the WS on both of these measures were significantly correlated. This very interesting study has several strengths, including a robust and well-matched sample, rigorous screening criteria, a balance of male and female participants, thorough measures of dietary compliance, a negative control memory task (hippocampal-independent memory test; Forward Digit Span), and a solid experimental design that allows for the ability to conclude that brief exposure to a WS *causes* deficits in both hippocampal-dependent cognitive function and appetitive control. Nonetheless, the report would be strengthened by addressing the following issues:

This research is guided, in part, by the premise that the hippocampus regulates intake through a process that requires hippocampal-dependent processing of interoceptive satiety cues. The authors cite a study (Henderson, Smith and Parent, 2013) as showing that hippocampal lesions impair use of internal state to resolve appetitive-related decisions (lines 51-52). That study does not appear to have results that support that interpretation. Although there is strong evidence that hippocampal damage does impair interoception, the authors should address recent findings demonstrating that hippocampal control of energy intake can occur independently of interoceptive satiety cues (Hannapel, R.L., et al 2019).

The authors appear to imply that their data demonstrate a causal relationship between hippocampal function and appetitive control and it seems as though they are suggesting that the fact that the correlation between the effects of the WS on hippocampal function and appetitive control is strong means that a causal relationship exists between the two variables. The title states that the WS impairs “hippocampal-dependent appetitive control” ; yet it is unclear how we know that it is hippocampal-dependent. Is there any evidence showing an association between confirmed hippocampal damage and reduced effect of satiety on liking and wanting in humans? Is there any possibility that there may be diet-induced dysfunction in brain areas involved in reward? There should be some serious consideration of the possible independence of the two measures, which may be supported, in part, by findings showing that hippocampal control of intake does not necessarily require interoception.

The face validity of the measure of appetitive control is limited and warrants discussion. Actual appetite control/eating behavior were not measured, but were instead assessed indirectly by a proxy dependent on self-report. Is there any evidence that this pattern of liking and wanting actually is correlated with and/or leads to overeating?

The finding that there were no differences in blood glucose levels following the WS breakfast compared to the control breakfast is surprising given that the WS breakfast contained a

percentage of sugar. Is there any possibility that the impairing effects of the WS diet were mediated by a deficiency in protein rather than by excess fat and/or sugar?

The want/like analyses and results are difficult to understand. The description of these results (line 320). "Group differences in changes in wanting vs. liking that accounted for this effect" is unclear. Also, the Figure depicting these results (Figure 1) is hard to follow. The data from the two diet groups are presented in separate figures even though the comparisons are between these two groups. It would be helpful if 1) there were separate figures for each dependent variable rather than for the two independent variables, 2) asterisks/symbols showing differences are added and 3) the figure legends explain what the findings show.

The finding that there are no differences between diet groups on the HVLTL at the 3-week follow up is interpreted to mean that the effect of the WS is reversible following discontinuation of the WS diet. However, the HVLTL scores in the WS group don't change between week 1 and week 3. The lack of effect of the WS seems to be actually due to the fact that control scores dropped from week 1 to week 3, so statements regarding recovery and reversibility appear to need to be tempered.

Appendix B

Dear Prof Viding,

Please find attached our revised MS entitled 'Hippocampal-dependent appetitive control is impaired by experimental exposure to a Western-style diet' which we would like to resubmit to *RS Open Science*. Before turning to the reviewer comments, we would like to thank all concerned for reading and commenting on our MS. The comments were very insightful and thought provoking, and have led we believe to an improved MS. Our response to reviewer comments are marked '>>' and are in boldface type.

Reviewer 1:

Stevenson et al examine the effect of one week Western style diet feeding on hippocampal dependent learning and memory (HDLM), and appetitive control, before and after the feeding intervention. They report that HDLM and appetitive control declined following the Western diet and suggest that Western diet has adverse neurocognitive effects in humans as observed in preclinical models.

It would be useful to provide the reader with a table showing the HVLT recall on learning trials, and the delayed recall (means =/- SD), as well as the %retention. It would also be useful to provide the norms for this test and address whether the effects of diet are statistically significant but fall within expected normal range for function.

>> A new Table 3 now presents the HVLT data for both groups on each of the test sessions. We agree it would be interesting to compare to normative data. Such data is published in Benedict et al., (1998; Clin Neuropsych., 12(1), 43-55). The issue with comparing our findings to these normative data, is that the most interesting comparison is for Day 8 - when our participants have already completed the test once. The published normative data is just for participants who had done the test once. This is important, because in the Benedict paper they also include test-retest data, and there is a clear improvement in performance between the two administrations, from 91% to 95% (%retention). While this test-retest data would appear the ideal for normative comparison, unfortunately these findings were obtained solely from elderly subjects. It is for these reasons that we have not been able to make normative comparisons.

Table 1 provides the Western Diet score for the two study groups which are 50.7 and 52.9. What is the average score for this test and how much below the average are these scores?

>> We now report in the Results, a brief statement regarding this matter. We have several years of data on students using our measure of Western dietary intake, and the meaning of our sample scores relative to this normative data is now described (please see ln324 onwards).

From the data presented in Figure 1 it appears that wanting of test foods in those consuming western diet was less before the meal, yet declined to the same rating after the meal. Thus it is not clear that this data supports subjects wanting more after finishing the meal? Please explain.

>> We feel two points are important to consider here. First, we now include an analysis of all the pre-breakfast ratings (please see line 380 onwards). This finds a general propensity for lower pre-breakfast ratings in the WS-diet group, crucially on all Days and on all Rating types. So prior to the intervention (and after), the WS-diet group reported slightly lower wanting, liking and want more ratings before breakfast. As this effect is present on both days and on all ratings, it cannot explain the key interaction effect, which results from a contrast of within-subject changes *across Days and Rating type* by Group. We have also now included additional analyses that further serve to clarify this effect (please see ln353 onwards) – as suggested by R 2. The second point we wish to make is that our key focus is on wanting relative to liking – put another way how much you *think* you'll like it versus how much you *actually* like it. It is this that changes in our study, and it is this that differed in our earlier study, as we outline in the Introduction (please see ln152 onwards).

Reviewer 2:

Comments to the Author(s)

Stevenson and colleagues examine the effects of a 1-week western-style diet (WD) intervention on hippocampal-dependent learning and memory (HDLM) and measures of 'wanting' and 'liking' of various foods. The WD intervention significantly reduces performance on a measure of HDLM but not on a digit-span task, replicating prior work by this group. To measure appetitive control participants rated how much they liked, wanted and wanted more of various foods before and after a breakfast. The authors compare pre- and post-breakfast change scores before and after the diet intervention, and claim that shifts in wanting vs. liking ratings on the day 8 test is evidence of weakened appetitive control produced by exposure to the WD.

The experiment is well-designed and the manuscript is clearly written. Results have interesting implications for the intersection of hippocampal-dependent learning and memory processes with acute appraisal to food reward value.

Major comments

1) The conclusion that HDLM performance recovered at follow up does not seem supported by the data. The absence of a Group effect on day 29 appears due to poorer performance in the control group (which decreased from 97 to 94.2%) rather than an increase in the WD group, whose performance was stable (92 to 92.4%). This conclusion should be revised.

>> This is a very fair point and was also made by R3. This claim has been removed and these results are now discussed from this perspective (please see ln469 onwards & abstract).

2) Could the differing energy content and macronutrient composition of the control and WD lab breakfasts confound interpretation of the pre-post changes on want/like measures? The WD breakfast produces smaller changes in hunger, fullness and thirst on day 1 and day 8, which could feasibly alter post-breakfast ratings. I wonder whether the proposed 'weakening of appetitive control' is not some artefact of meal composition. This could be discussed.

>> This is also a very fair point and this possibility is now discussed (please see ln497 onwards).

3) The results of the wanting and liking test are somewhat difficult to interpret. The 3-way interaction found for the wanting/liking test, and subsequent interaction with only wanting vs. liking, gives good evidence for WD effects. However these analyses are examining changes in change scores, and even the 'follow-up' interaction appears to have 4 factors: wanting vs. liking, pre vs. post, WD vs. control, day 1 vs. day 8. As such it still isn't clear what drives the interaction, and I think further analyses would help bolster the authors' case. Some suggestions (but others may be more appropriate):

a. Within the WD group, does the pre-post change in wanting vs liking change from day 1 to day 8? I.e., directly test the hypothesis that the drop in liking becomes more pronounced only in WD participants.

b. Considering only day 8 data, is the group x scale (want vs. like) interaction significant?

c. Comparing pre-meal measures between groups on day 1 vs. 8. The WD group appears to make lower pre-meal ratings on all 3 subscales on both days, particularly on day 8 (although here there might be reason to expect group differences). Was this significant?

d. Showing the significant difference graphically would greatly help the reader interpret the figure.

>> R3 & R1 also suggested revisions here. We have now revised and added to the analyses of these data in the Results as suggested (please see ln343 and onwards), revised the bottom panels of Figure 1 to more accurately reflect the key interaction effect and amended the figure legend to more clearly point out the central finding.

4) Digit span: performance seems uncannily high on this task (13-15 numbers recalled) – do the means refer to the number of digits recalled?

>> These are scores on the digit span task, rather than digit spans *per se*. We did not make this clear. We have now clarified this in the Method (ln 267 onwards) and used the phase digit span score in the Results rather than digit span.

5) Lines 341-348: this analysis looks interesting, and the correlations seem robust, but which change score is being evaluated – wanting, liking, or wanting more? Only one rho value is provided.

>> This section also needed significant clarification. We have now provided a far more detailed description of the scores used in this analysis and hence why the initial test just involves one Spearman's rho (please see ln429 onwards).

6) In several instances non-significant results are dismissed without further comment. Such results still hold great value and may be of interest to those in the field (e.g. there is a growing field examining diet effects on mood). Adding even descriptive data (in supplementary form) would permit their inclusion in meta-analyses and would be a valuable asset.

>> **We agree. In the supplementary data file we have now included the DASS scores from Day 1, 8 and 29, along with the mood and alertness scores as well. This is flagged now in the MS.**

Minor comments

Line 74: suggest changing 'a week' to 'four days'.

>> **Completed.**

Line 112-115: this sentence was difficult to follow – suggest splitting and/or rephrasing.

>> **Revised (please see ln120 onwards)**

Line 183: please define what is meant by 'below average' – did this refer to the average (mean/median?) of the present sample, or to reference data from Francis & Stevenson, 2013?

>> **This point was also raised by R 1. In response we have now provided reference data to put the DFS values in context (please see ln324 onwards).**

Line 201: what instructions were given to the WS-diet group regarding their other meals?

>> **They were instructed that apart from the requested dietary changes, they were to try and otherwise eat as they normally would (please see ln229).**

Line 212: Is it true that the control breakfast was 58% protein? Could protein and carbohydrate proportions have been swapped mistakenly?

>> **These values are correct, as protein shake powder was a principal ingredient of the thickshake, with low fat meat and cheese in the toastie.**

Line 215: Please define 'Mini-toast'. 'Froot', not Fruit loops (regrettably) is the spelling

>> **Apologies for the misspelling of froot loops (corrected) and mini-toasts are now defined (please see ln236 onwards).**

Line 248: word missing, 'how much more'

>> **Corrected.**

Lines 321-323: Please add the results of these statistical tests

Line 443: typo, 'Atuquayefio'

>> **Corrected.**

Line 454: it could be added that altered BBB permeability tends to be observed only after long diet exposures (i.e., after cognitive impairment).

>> **This has now been modified accordingly.**

Reviewer 3:

This study investigated whether a high fat/high sugar (AKA Western; WS) diet impairs hippocampal-dependent learning and memory and appetitive control in humans as it does in rodents. Healthy lean adults were randomized to a 1-week WS diet or continued with their habitual diet. Hippocampal function was measured using the Hopkins Verbal Learning Test; HVLTL) and appetitive control was assessed using ratings of liking and wanting of palatable snack foods under conditions of hunger and satiety. Compared to the control/habitual diet, the WS impaired HVLTL performance and decreased the effects of satiety on wanting/liking, and the effects of the WS on both of these measures were significantly correlated. This very interesting study has several strengths, including a robust and well-matched sample, rigorous screening criteria, a balance of male and female participants, thorough measures of dietary compliance, a negative control memory task (hippocampal-independent memory test;

Forward Digit Span), and a solid experimental design that allows for the ability to conclude that brief exposure to a WS causes deficits in both hippocampal-dependent cognitive function and appetitive control. Nonetheless, the report would be strengthened by addressing the following issues:

This research is guided, in part, by the premise that the hippocampus regulates intake through a process that requires hippocampal-dependent processing of interoceptive satiety cues. The authors cite a study (Henderson, Smith and Parent, 2013) as showing that hippocampal lesions impair use of internal state to resolve appetitive-related decisions (lines 51-52). That study does not appear to have results that support that interpretation. Although there is strong evidence that hippocampal damage does impair interoception, the authors should address recent findings demonstrating that hippocampal control of energy intake can occur independently of interoceptive satiety cues (Hannapel, R.L., et al 2019).

>> The Henderson study has been deleted accordingly. The Hannapel study was not known to us and we thank the reviewer for pointing it out. We now cite this study in the Introduction and Discussion, noting that inhibitory effects can occur in animals independent of state-based changes and its potential implications (please see paras starting on ln113, ln528 and ln621).

The authors appear to imply that their data demonstrate a causal relationship between hippocampal function and appetitive control and it seems as though they are suggesting that the fact that the correlation between the effects of the WS on hippocampal function and appetitive control is strong means that a causal relationship exists between the two variables. The title states that the WS impairs “hippocampal-dependent appetitive control” ; yet it is unclear how we know that it is hippocampal-dependent. Is there any evidence showing an association between confirmed hippocampal damage and reduced effect of satiety on liking and wanting in humans? Is there any possibility that there may be diet-induced dysfunction in brain areas involved in reward? There should be some serious consideration of the possible independence of the two measures, which may be supported, in part, by findings showing that hippocampal control of intake does not necessarily require interoception.

>> We agree that this is an important issue. Consequently, we now examine it in the Discussion (please see para starting on ln621).

The face validity of the measure of appetitive control is limited and warrants discussion. Actual appetite control/eating behavior were not measured, but were instead assessed indirectly by a proxy dependent on self-report. Is there any evidence that this pattern of liking and wanting actually is correlated with and/or leads to overeating?

>> This too is an important point, and again we now consider this in the Discussion (please see para starting on ln515).

The finding that there were no differences in blood glucose levels following the WS breakfast compared to the control breakfast is surprising given that the WS breakfast contained a percentage of sugar. Is there any possibility that the impairing effects of the WS diet were mediated by a deficiency in protein rather than by excess fat and/or sugar?

>> These points are now examined in the Discussion (please see para starting on ln607)

The want/like analyses and results are difficult to understand. The description of these results (line 320). “Group differences in changes in wanting vs. liking that accounted for this effect” is unclear. Also, the Figure depicting these results (Figure 1) is hard to follow. The data from the two diet groups are presented in separate figures even though the comparisons are between these two groups. It would be helpful if 1) there were separate figures for each dependent variable rather than for the two independent variables, 2) asterisks/symbols showing differences are added and 3) the figure legends explain what the findings show.

>> Similar points were made by R2. We have now revised and added to the analyses of these data in the Results (please see ln343 and onwards), revised the bottom panels of Figure 1 to more accurately reflect the key interaction effect and amended the figure legend to more clearly point out the central finding.

The finding that there are no differences between diet groups on the HVLТ at the 3-week follow up is interpreted to mean that the effect of the WS is reversible following discontinuation of the WS diet. However, the HVLТ scores in the WS group don't change between week 1 and week 3. The lack of effect of the WS seems to be actually due to the fact that control scores dropped from week 1 to week 3, so statements regarding recovery and reversibility appear to need to be tempered.

>> This is a very fair point and was also made by R2. This claim has been removed and these results are now discussed from this perspective (please see ln469 onwards & abstract).

In conclusion, we again thank the editors and referees for reading and commenting on the MS.

Yours sincerely,

Dick Stevenson, DPhil
12/11/19

Appendix C

Dear Prof Viding,

Please find attached our revised MS entitled 'Hippocampal-dependent appetitive control is impaired by experimental exposure to a Western-style diet' for *RS Open Science*. We were asked to make a few minor changes by R2 as well as making sure that the end statements were in order. Our responses are in bold and follow a '>>'.

Associate Editor Comments to Author:

Overall, the reviewers seem happy with your changes, but have a couple of minor final suggestions for the presentation of the paper.

>> We thank the associate editor for these comments and note that the end order statements are now all in order.

Reviewer comments to Author:

Reviewer: 2

Comments to the Author(s)

Symbols indicating the significant group differences on day 8 could be added to Table 3.

>> We did not add symbols indicating group differences. There were two reasons for this. First, the tabled data was added at the request of the reviewers to show readers all of the response data from which the key results were derived. However, the response data itself was not analysed - and so statistical comparisons are not available. Second, the only data analysed (i.e., the key results) was percent retention, but the way in which this was analysed does not readily translate to the way in which the data are tabled (i.e., we used ANCOVA, and so the key comparisons are adjusted means as reported in the body text of the Results). It was for these reasons that we chose not to add significance symbols.

The figure legends on the bottom row of Fig 1 are overlapping.

>> Thanks for spotting this. We are not sure how this arose as it is not present in the original word document nor when we make a pdf file from this document. Nonetheless, we have re-made the figure and hopefully this should sort out this problem.

Reviewer: 1

Comments to the Author(s)

Reviewer thanks the authors for consideration and response to prior critique. No further comments/questions.

>> We thank the Reviewer for these comments.

Yours sincerely,
Dick Stevenson, DPhil
6/1/20